# Rare Forms of Lipomatosis: Dercum’s Disease and Roch-Leri Mesosomatous Lipomatosis

**DOI:** 10.3390/jcm10061292

**Published:** 2021-03-21

**Authors:** Madleen Lemaitre, Sebastien Aubert, Benjamin Chevalier, Arnaud Jannin, Julien Bourry, Gaetan Prévost, Herve Lefebvre, Marie-Christine Vantyghem

**Affiliations:** 1Endocrinology, Diabetology and Metabolism, CHU Lille, 59000 Lille, France; benjamin.chevalier@chru-lille.fr (B.C.); arnaud.jannin@chru-lille.fr (A.J.); julien.bourry@chru-lille.fr (J.B.); 2U1190, EGID European Genomic Institute for Diabetes, University Lille, 59000 Lille, France; 3Department of Biopathology, CHU Lille, 59000 Lille, France; Sebastien.AUBERT@CHRU-LILLE.FR; 4Department of Endocrinology, Diabetes and Metabolic Diseases, Rouen University Hospital, 76000 Rouen, France; gaetan.prevost@chu-rouen.fr (G.P.); herve.lefebvre@chu-rouen.fr (H.L.); 5Inserm, U1190, 59000 Lille, France

**Keywords:** lipodystrophy, Dercum’s disease, Roch-Leri mesosomatous lipomatosis, lipomatosis, adipose tissue

## Abstract

In contrast to obesity, which is very frequent, lipomatosis and lipodystrophy syndromes are rare diseases of adipose tissue. Lipodystrophy syndromes are characterized by metabolic abnormalities associated with partial or generalized lipoatrophy. Lipomatosis is defined by the presence of several body lipomas without lipoatrophy. Dercum’s disease (DD) and Roch-Leri mesosomatous lipomatosis (RLML) are rare and poorly characterized forms of lipomatosis. They have raised little clinical interest despite the non-negligible consequences of DD on quality of life. The main clinical presentation of these diseases includes multiple lipomas, which are painful in DD (in contrast to RLML). The two diseases are frequently associated with obesity and metabolic syndrome, with hypertension, diabetes, or dyslipidemia. The long-term course of the diseases remains poorly described. DD affects mainly women, whereas RLML mostly affects men. In both diseases lipomas are found on the back and thighs, as well as on the abdomen in DD and the forearms in RLML. The painful lipomas tend to recur after surgery in DD (in contrast to RLML). Most cases are sporadic. No specific treatment has been identified, as the pathophysiology remains unknown. Nevertheless, low-grade fat inflammation and specific abnormalities such as hyperbasophilia deserve further investigation. The aim of this review is to analyze the available literature on the topic.

## 1. Introduction

In recent decades there has been renewed interest in adipose tissue, and various diseases relating to its development have been reported. In contrast to common obesity, lipomatosis and lipodystrophy syndromes are rare [1]. Lipodystrophy syndromes are characterized by metabolic abnormalities such as insulin resistance, hypertriglyceridemia, and fatty liver due to the limited ability of the subcutaneous adipose tissue to store triglycerides. They are associated with partial or generalized lipoatrophy, a hallmark of these syndromes [2]. Lipomatosis is defined by the presence of several lipomas in the body. Lipomas may cause discomfort and are unaesthetic in appearance. The absence of the lipoatrophic component is one of the main differences between lipomatosis and lipodystrophy syndrome. Nevertheless, lipomatosis is sometimes included in lipodystrophy syndrome classifications. Lipomatosis can be an isolated manifestation or may also be part of a syndrome. Different entities have been described (Table 1) [3].

Adipose tissue is an endocrine organ that includes different components, especially white and brown adipose tissue (both of which are active metabolically). White adipose tissue (WAT) stores excess energy in the form of triacylglycerols, and brown adipose tissue (BAT) dissipates energy in the form of heat. WAT has the capacity to secrete numerous adipokines (leptin, adiponectin, Tumor Necrosis Factor-alpha (TNF-alpha), etc.), which allows the control of energy metabolism but also the regulation of the immuno-inflammatory response. Subcutaneous adipose tissue (SAT) contains many immune cells such as monocytes/macrophages, mast cells, and lymphocytes, which participate in hormonal regulation. The distribution of fat between the visceral, subcutaneous, and medullary bone components, and between the upper and lower parts of the body, regulates the insulin resistance level, as do environmental factors. Obesity is then associated with chronic low-grade inflammation, which regulates innate and acquired immunity within the SAT itself [4,5].

Lipomatosis is frequently associated with obesity [6]. The rarity of most types of lipomatosis probably explains their poor description in the literature. They are generally considered to be benign, but some have a non-negligible impact on the quality of life. Few longitudinal studies have been published in this regard. The objective of this review is to present the available data on two of these forms of lipomatosis: Dercum’s disease and Roch-Leri lipomatosis.

## 2. Dercum’s Disease

Dercum’s disease is an orphan disease that is listed by Orphanet (ORPHA:36397) and the National Organization for Rare Disorders (NORD) [7]. It is characterized by the progressive and insidious appearance of multiple lipomas, which result in chronic pain (>3 months) and have deleterious consequences on quality of life [8]. Epidemiologically, the disease preferentially affects female adults (sex ratio: 5–10 females to one male), generally after menopause, although all ages can be affected. The prevalence of the disease has not yet been defined except in a Swedish population, where it was found to be 1/12,000 [9]. Dercum’s disease has many synonyms: adiposis dolorosa or juxta-articular adiposis dolorosa, fatty tissue rheumatism, painful lipomatosis of morbus Dercum, adiposalgia, or Ander syndrome.

### 2.1. History

Dercum’s disease was first described in 1888 by François-Xavier Dercum (1856–1931), then a distinguished American neurologist at the University of Pennsylvania. It was initially named “adiposis dolorosa”, a name which is no longer used [10]. The two symptoms that enabled Dr. Dercum to define the disease were the combination of adipose tissue pain and weight gain [11].

### 2.2. Pathophysiology

The pathophysiology of Dercum’s disease is not yet understood. The painful component cannot be explained by abnormalities in the fat tissue growth. Multiple hypotheses have been proposed, including compression of the neighboring nerves by lipomas. Other hypotheses include an autoimmune origin, an endocrine abnormality such as thyroid dysfunction (as proposed by Dr. Dercum), pituitary dysfunction [12], or steroid therapy [13]. Further suggestions, although without evidence, include inflammatory causes (such as rheumatoid arthritis) [14] or infections [15], alcohol, traumatism [16], microvascular thrombi, lack of synthesis of long-chain fatty acids [17], impaired response to insulin or norepinephrine [18], sudden weight change as suspected in a case of metabolic bypass surgery [19], and, finally, autonomous nervous system dysfunction. More recent studies have raised the possibility of an abnormal lymphatic system, since patients with Dercum’s disease show hypertrophied, tender, fibrotic, and tubular lymphatic tissue within the adipose tissue, which is fluorescent in near-infrared fluorescence lymphatic imaging (NIRFLI) [20]. An abnormal communication between adipose and lymphatic tissues would therefore be at the origin of DD, as suggested by Dr. Dercum himself after the autopsy of a DD case [21].

The disease is usually sporadic. However, some cases could be autosomal dominant with incomplete penetrance [22,23,24].

### 2.3. Symptomatology

The main clinical symptoms include painful lipomas in overweight or obese people. Obesity sometimes occurs in the absence of poor lifestyle choices, and optimization of lifestyle does not always lead to successful weight loss [25]. Even if it is successful, it does not result in reduced pain (personal communication). This pain is usually chronic, debilitating, and resistant to conventional analgesics; there is increased exquisite tenderness with deep palpation of the lipomas. Mild irritation can trigger the pain (e.g., temperature changes, rubbing of clothes) [26]. It is located mainly on the trunk, the proximal extremities of the limbs, the buttocks, and the pelvis (Figure 1). Lipomas are very rarely found on the head or neck (a differential diagnosis can be made with Launois-Bensaude syndrome, in which lipomas are not painful). Multiple classifications have been proposed over time by Guidice Andrea ((1900)—Type I: nodular type; Type II: diffuse type; Type III: mixed type), Roux and Vitaut (1901), Stern (1910), Labbé and Boulin (1927), Gram (1930), Herbst (2007), and Hansson (2012). The current classification of Dercum’s disease according to the Orphanet Journal of Rare Diseases (OJRD) refers to four subtypes:***Type*** ***I*:**Generalized diffuse form defining a diffuse pain component without clear lipomas.***Type*** ***II*:**Nodular form corresponding to generalized pain but of greater intensity in and around lipomas.***Type*** ***III*:**Localized nodular form characterized by pain in and around several lipomas.***Type*** ***IV*:**Juxta-articular form corresponding to solitary deposits of excess fat (example: medial aspect of the knee) [27].

Accompanying signs such as vascular fragility (ecchymotic tendency) and neuropsychic disorders (anxiety, sleep disorders, memory impairment, and depression (as seen in 55% of a series of 111 women with Dercum’s disease [28])) are frequent. Less specific disorders such as digestive symptoms (constipation, bloating, and chronic abdominal pain), tachycardia, shortness of breath, asthenia, generalized muscle weakness, arthralgia [29], and repeated infections (vaginitis) have been reported. An exceptional case of septic shock secondary to steatonecrosis of a lipoma [30] and lipomatous hypertrophy of the interatrial septum [31] has been described. Mastalgia may exceptionally be a hallmark of the disease [32], as well as headache related to lipomas of the scalp [33].

### 2.4. Biological Parameters

There is no specific biomarker for Dercum’s disease. Metabolic syndrome is often present and different immunoinflammatory markers have been studied. 

On the metabolic level, the presence of larger-than-normal particles of High Density Lipoprotein (HDL)-c3 (which are less cardioprotective than HDL-2), as well as an increase in Lp-PLA2 (phospholipase A2 associated with lipoproteins) were found in one pediatric case and were associated with hyperinsulinism [34]. Other case reports have described hypercholesterolemia in adults with Dercum’s disease [35,36]. However, high cholesterol can also be secondary to being overweight, and the specific association with Dercum’s disease itself remains to be proven. In another study, the lipoprotein lipase activity of seven DD patients was lower than the usual reference, with this result tending towards significance [37].

A study of 94 DD patients showed the presence of diabetes in 16% and hypertension in 21% of cases. These frequencies were not statistically different when compared with a group of 160 patients with lipedema. Hypothyroidism was the most common past medical condition and was present in 26% of DD patients and 27% of the lipedema group [38].

Modulation of the neuropeptides involved in the pain signal was demonstrated in the serum and cerebrospinal fluid (CSF) of 53 DD patients with respect to substance P, neuropeptide Y, and ß-endorphin. In the CSF, substance P and neuropeptide Y levels were lower, and the ß-endorphin level was higher than in the obese control group [28]. Of note is that these three substances are involved in food intake and are associated with obesity [39,40]. Furthermore, endocannabinoid receptors of subcutaneous fat are thought to be dysregulated in obesity, thereby inducing inflammation [41]. They were not specifically studied in DD despite the fact that they could perhaps play a role in the pain component of the disease.

With regard to inflammation, different immunoinflammatory parameters have been studied. An increase in the blood sedimentation rate and the serum level of complement total (CH50), which plays a role in macrophage-adipose tissue communication, has been demonstrated. Certain blood cytokines, such as macrophage inflammatory protein (MIP)-1b and fractalkine were found to be lower in 10 DD patients as compared to 5 healthy control subjects, in association with significantly higher levels of interleukin (IL)-13 in DD subjects. Higher IL-13 levels could downregulate fractalkine and MIP-1b through suppression of lipopolysaccharides (LPS)-induced production of proinflammatory cytokines. There were no significant differences for other cytokines or adipokines (leptin, adiponectin, plasminogen activator inhibitor-1, IL-1b, IL-6, IL-8, IL-10, MIP-1a, MCP-1, or TNF-a) [42].

### 2.5. Diagnosis

Dercum’s disease is a clinical diagnosis that is usually made by specialists in the study of adipose tissue as well as internists, dermatologists, or plastic surgeons. In 2012, after Dercum, Hansson formulated two main diagnostic criteria: obesity and chronic pain in hyperplasic/hypertrophic adipose tissue [8]. Misdiagnosis is common in most cases before the final diagnosis is made. Different differential diagnoses need to be ruled out:

**Lipoedema**, which is suggested by an adipose tissue abnormality with a cuffing or “bracelet” effect to the wrists and ankles. Painful lipomas are not present. The hips and thighs may show a saddle bag appearance with large skin folds. Higher tissue water levels in lipedema than in Dercum’s disease could help to distinguish these two diseases [43]. Recently, lipedema has been presented as a potential estrogen-dependent adipose tissue disorder that could be triggered by caveolin-1 (CAV1) dysfunction [44].

**Fibromyalgia** is characterized by muscle pain concomitant with recurrent episodes of intense fatigue. The diagnosis is confirmed only in the presence of [45]:-Painful symptoms for at least three months,-A Widespread Pain Index (WPI) of 7 and a Symptom Severity (SS) scale score of 5, or WPI between 3 and 6 and SS scale score of 9,-The elimination of any other cause of chronic osteoarticular pain.

**Panniculitis** is the inflammation of the subcutaneous adipose tissue, which is characterized by tender skin nodules and systemic signs. The diagnosis is made after skin biopsy to determine the septal or lobular features and the presence and type of subcutaneous inflammatory infiltrates (neutrophils, lymphocytes, histiocytes, granulomas) [46].

**Madelung disease**, also known as Launois-Bensaude syndrome, is a form of multiple symmetric lipomatosis that is mainly localized in the upper body (shoulders, neck, head) and occurs preferentially in men with chronic excessive alcohol consumption. Multiple deletions of mitochondrial DNA, *MERRF*, *LIPE*, and *MNF2* mutations have been identified in non-alcoholic cases [47]. The mutation described in Launois-Bensaude syndrome (mitochondrial DNA) has never been reported in cases of Dercum’s disease [20].

**Familial multiple lipomatosis** corresponds to the occurrence of painless lipomas mainly in the arms and legs from adolescence [48]. Autosomal dominant inheritance has been reported but the penetrance is variable within the same family. Abnormalities of the 12q13-15 region, which contains the *HMGA2* gene and *PALB2* gene mutation, have been reported [49]. 

**Type I neurofibromatosis**. Neurofibromas related to mutations in the *NF1* gene may be a differential diagnosis for lipomas. They are generally painless, except in case of degeneration [50].

**MERRF** (myoclonic epilepsy with red ragged fibers) is an extremely rare mitochondrial disease characterized by myoclonic episodes, cerebellar ataxia, and myopathy. Multiple lipomatosis is associated in rare cases [51].

**Circumscribed mesosomatous lipomatosis**, also called Roch-Leri lipomatosis, is a form of non-painful benign lipomatosis (see below).

**Proteus syndrome** is a rare disorder characterized by disproportionate, asymmetric overgrowth of skin, bones, blood vessels, and fatty and connective tissue, related to a mosaic activating mutation in the *AKT1* oncogene [52].

**CLOVES syndrome** is a congenital pathology associating asymmetric lipomatous overgrowth, vascular malformation, epidermal nevi, scoliosis, and skeletal and spinal anomalies [53]. 

**HHML syndrome** (hemihyperplasia–multiple lipomatosis syndrome) is a rare, genetic overgrowth syndrome (progressive, asymmetrical, moderate hemihyperplasia) associated with multiple, slow-growing, painless, subcutaneous lipomas distributed throughout entire body [54].

**PTEN hamartoma tumor syndrome (PHTS**) is a spectrum of disorders caused by mutations of the *PTEN* tumor suppressor gene characterized by multiple hamartomas, like SOLAMEN syndrome which includes segmental overgrowth, lipomatosis, arteriovenous malformation, and epidermal nevus [55].

**Gardner syndrome**, related to *APC* gene mutations, is a form of familial adenomatous polyposis characterized by multiple digestive adenomas (colon and rectum) associated with osteomas and multiple skin and soft tissue tumors, including lipomas [56].

**Psychiatric disorders**, when the pain complaint is not associated with objective disease.

**Endocrinopathies**, especially Cushing syndrome, are often associated with facial-troncular fat accumulation and hypothyroidism.

**Multiple endocrine neoplasia type 1**, related to a *MEN1* gene mutation, is often associated with lipomas and sometimes hibernomas (lipoma of brown adipose tissue), epidural lipomatosis, and familial angiolipomatosis, for which the prevalence is unknown.

**Atypical lipomatous tumors** have been defined as well-differentiated liposarcomas exhibiting a higher frequency of local recurrence after surgical resection. They are generally >4 cm in size and exhibit hardness, poor tumor mobility, and a septal structure on MRI [57].

### 2.6. Paraclinical Investigations

Morphologically, ultrasound evaluation showing the presence of multiple oblong lipomas in the subcutaneous adipose tissue with a larger axis measuring less than 2 cm and a hyperechoic nature may help to confirm the diagnosis of lipoma if there is doubt concerning a malignancy. In fact, distinguishing a lipoma from a well-differentiated liposarcoma is complex on ultrasound; the behavior of the tumor to compression by a transducer may be a differential criterion. For the most aggressive forms, an uneven delineation, a heterogeneous appearance, infiltration in the adjacent tissues, and pathological blood flow may support the diagnosis of a liposarcoma. Myxoid liposarcoma is distinguished by a cystic structure [58]. 

On computerized tomography (CT) scan, lipomas are predominantly stalked or sessile within edematous subcutaneous fatty tissue [59]. MRI does not show edema or inflammation but an increased signal (“blush-like” effect) in Dercum’s disease [60].

### 2.7. Histological Analysis

Histological analysis of lipomas can confirm the lipomatous and benign nature of the tissue if there is any doubt concerning malignancy. Nevertheless, histologic examination does not always reliably distinguish benign lipomas from atypical lipomatous tumors/well-differentiated liposarcomas, nor dedifferentiated liposarcomas from other pleomorphic sarcomas, which are entities with different prognoses and management. Molecular confirmation of pathognomonic 12q13-15 amplifications leading to MDM2 overexpression in sarcomas is a diagnostic gold standard in the case of doubt [61].

There are a few histological data regarding Dercum’s disease (Table 2) [37,42,62,63,64]. These histological results suggest that adipose tissue in Dercum’s disease might be less metabolically active due to the preponderance of connective tissue and fibrosis. The presence of multi-nucleated giant cells produced by activated, pro-inflammatory macrophages was identified in DD adipose tissue, which suggests their role in adipose tissue accumulation and in resistance to weight loss [42]. An inflammatory response was also shown in the adipose tissue from another DD series. However, this response was not more pronounced than that observed in healthy obese controls, thereby suggesting an inflammatory component linked to obesity and not directly to Dercum’s disease [64].

### 2.8. Prognosis

Few studies have been done with regard to the natural course of the disease; however, the tendency seems to be towards worsening symptoms over time [65].

### 2.9. Therapeutic Management

While there is no formal diagnostic marker for Dercum’s disease, there is likewise no cure. The symptomatic treatment of pain and associated comorbidities must be adapted to each patient. The different treatments proposed in the literature are mainly reported as case reports or short surgical series without control groups [66].

#### 2.9.1. Analgesic Treatment

Lidocaine is administered in the form of transdermal [67] or intravenous injections as an adjunct treatment that cannot be used over the long term given the risk of adverse effects. Mexiletine is an oral lidocaine analogue with a longer metabolism which also appears to have good efficiency but requires prior cardiologic assessment [68]. Ketamine infusions may be an alternative [66]. 

Level I analgesics seem to have little or no effect. However, recent data indicate good efficacy of nonsteroidal anti-inflammatory drugs (NSAIDs) (level II) or opiates (level III); both classes of drugs have specific side effects, particularly renal toxicity for NSAIDs and dependency for opiates. Calcium chain modulators such as pregabalin or oxcarbazepine, also used as antidepressants, may provide efficient pain management [69,70]. Antidepressants acting via inhibition of serotonin reuptake have not be specifically studied in DD but could be of particular interest taking into account the high serotonin levels associated with obesity (see Section 4). 

#### 2.9.2. Psychological Support

Psychotherapy sessions and stress management aim at limiting exposure to any source of psychological or physical stress that could lead to an upsurge in pain symptoms.

#### 2.9.3. Surgery

Surgery may be of use in the subgroup of patients for which pharmacotherapy does not provide a benefit or who have limitations in their mobility.

Liposuction is only conceivable for extremely severe chronic pain that impacts the activities of daily living or for diffuse pain throughout the adipose tissue (type I and type II). The results appear to show a reduction in pain [71]. The improvement in quality of life is, however, only moderate and transient, and remains inferior to that of control subjects [72].

The excision of lipomas is a technique that has mainly been proposed in type III and type IV adiposis dolorosa but is now rarely performed due to the risk of lipoma recurrence after surgery. Nevertheless, it allows for mobility to be regained and a temporary reduction in pain in type IV forms without major postoperative complications, except for multiple scars [27]. Recently, a minimal incision technique was proposed for type III forms [73].

#### 2.9.4. Treatment of the Metabolic Syndrome

Lifestyle measures including exercise do not influence the disease directly but result in better overall well-being, the benefits of which cannot be neglected in the context of this chronic disease. However, some case reports mention no improvement or a worsening of symptomology with weight loss [19,25].

Metformin was found to improve pain symptoms in one patient with Dercum’s disease who had been newly diagnosed with type 2 diabetes [74].

#### 2.9.5. Other Therapies Reported in the Literature (Only in Case Reports)

Pharmacological therapies: Methotrexate and infliximab [75], interferon alpha 2-b [76], and corticosteroids have been proposed. Immunosuppressants such as azathioprine, cyclosporine, hydroxychloroquine, etc., have been reported as effective in refractory forms, but there is only one report in literature [77].

Conservative non-surgical therapies such as rapid cycling hypobaric pressure [78], electrocutaneous stimulation using FREMS (frequency rhythmic electrical modulation system) and MC5-A Calmare [79], or manual deep massage in lipomatous areas reduce masses, nodules, and the amount of tissue fluid [80].

Coolsculpting, a non-invasive procedure that uses cryolipolysis to induce lipolysis without damaging the surrounding tissue, could successfully treat painful abdominal lipomas (one case report) [81].

The subcutaneous perineural injection of a dextrose solution (prolotherapy) could reduce neuropathic inflammation but is not recognized in every country [66].

Acupuncture has also been proposed.

To conclude in this first part, Dercum’s disease, a painful form of lipomatosis often associated with obesity, was first described more than 130 years ago. Despite its consequences with regard to quality of life, it remains an orphan disease in terms of clinical and biological markers, pathophysiology, and treatment. Nevertheless, advances in the pathophysiology of adipose tissue and in the genetic causes of lipodystrophy syndromes have raised new interest in this rare form of lipomatosis.

## 3. Roch-Leri Lipomatosis or Lesosomatous Lipomatosis

Roch-Leri or mesosomatous lipomatosis (RLML) is an orphan disease listed by Orphanet (ORPHA:529) [82] and by the National Organization for Rare Disorders (NORD). This disorder corresponds to rare and benign fatty tissue overgrowth characterized by the presence of numerous small lipomas located on the middle third of the body. The denomination comes from the Greek “mesos” (middle) and “soma” (body). In contrast to Dercum’s disease, this form of lipomatosis is painless. Epidemiologically, the disease preferentially affects male adults in the third decade of life, although all ages can be affected [83]. The prevalence of the disease has not yet been defined. Throughout the literature, synonyms include mesosomatic, discrete, or Roch lipomatosis and multiple circumscribed mesosomatic lipomatosis.

### 3.1. History

There are currently fewer than 15 articles in the literature referring to this rare form of lipomatosis. In addition, there is confusion in many articles between benign multiple lipomatosis (Madelung disease), familial multiple lipomatosis, and RLML. The first description of the disease was provided by Brodie in 1846. However, its discovery is attributed to Dr. Roch, a Swiss internist (1878–1967), and to Dr. Léri, a French physician (1875–1930) who described a distinct nosological entity under the name “discrete lipomatosis” [84].

### 3.2. Pathophysiology

The pathophysiology remains unknown. The disease appears to be mostly sporadic. However, some cases could be autosomal dominant [85].

### 3.3. Symptomatology

The disease is clinically characterized by the progressive occurrence of nodules in the subcutaneous adipose tissue. These lipomas measure 2 to 5 cm in diameter, are painless, multiple, asymmetrical, and are mainly located on the trunk and forearms (Figure 2). An exceptional subconjunctival localization has been reported [86]. No other symptoms are associated, although a few cases of thrombocytopenic purpura have been reported [87]. No complications of this disease have been mentioned.

### 3.4. Biological Parameters

No data are currently available in the literature.

### 3.5. Diagnosis

The diagnosis of RLML is clinical and is made after ruling out other diagnoses since there is no specific criteria or marker for this disease. The diagnosis is often ignored due to lack of questioning in patients who show no accompanying signs. Only patients who have an esthetic complaint consult a specialized center. Differential diagnoses that should be excluded are:

**Neurofibromatosis type 1**, related to NF1 gene mutation [50];

**Legius syndrome**, also known as NF1-like syndrome, which is a rare, genetic skin pigmentation disorder characterized by multiple café-au-lait macules with or without axillary or inguinal freckling [88];

Other types of lipomatosis (see Section 2.5).

### 3.6. Paraclinical Investigations

No study has characterized the biological and morphological aspects of patients with RLML. Ultrasound examination is the first line of investigation to confirm the fatty nature and benign characteristics of the subcutaneous nodules; lipomas appear as soft, variably echogenic masses. If encapsulated, the capsule may be difficult to identify on ultrasound [89].

### 3.7. Histological Analysis

There are no currently available data concerning the histological study of lipomas from patients with RLML. 

### 3.8. Therapeutic Management

Treatment is symptomatic and mainly based on surgery for these lesions when they cause discomfort, especially for esthetic purposes. Surgery is performed on non-diffuse forms of the disease. Lipomas usually do not recur after surgery. Weight loss can cause lipomas to reduce in size (personal experience).

To conclude this second part, Roch-Leri mesosomatous lipomatosis is a form of benign painless lipomatosis. The pathophysiology is unknown, and the disease appears to be somewhat forgotten. 

## 4. Contribution to Pathophysiology in Dercum’s and Roch-Leri Diseases

Dercum’s disease (DD) and RLML are rare and poorly characterized diseases. Their clinical presentations involve multiple lipomas, which are painful in DD as opposed RLML, without lipoatrophy. These two mild disorders have raised little interest in past decades. 

Nevertheless, recent discoveries regarding the different types of adipose tissue (white, brown, beige), its ability to secrete hormones and inflammatory mediators, its crosstalk with all human organs, and the consequences of its inflammation have currently led to renewed interest in these rare diseases. These advances can offer new insights into the physiology and diseases of adipose tissue. Lipomatoses have been recently included in the registry of the European Consortium for Lipodystrophy [90]. As of today, more than 70 genes have been identified in disorders of adipose tissue since the first identification of *LMNA* mutations in Familial Partial Lipodystrophy Syndrome type 2 (FPLD2) in the year 2000 [91]. New genes have been recently identified in lipoedema [92,93] and in some kinds of lipomatosis [94], but not in DD or RLML. Nevertheless, few series of DD or RLML disease are available. We recently analyzed the metabolic and inflammatory characteristics of 9 DD and 11 RLML patients, as well as 18 healthy controls who were included in a single-center cohort (NCT: 01784289). DD and RLML were nearly always associated with obesity and metabolic syndrome. Females were more frequent in the DD group and males in the RLML group, as already reported in the literature. The median age of the patients was similar in the three groups and was younger than that usually reported for DD (post-menopausal), which argues for a specific rather than an unspecific mechanism. Protein C (CRP) levels tended to be higher in the lipomatosis groups than in the control group, with basophilia in the DD group (doi:10.1210/jendso/bvaa046.785). 

Numerous data have demonstrated the link between obesity, the metabolic phenotype, and the inflammatory profile of adipose tissue. In obese adipose tissue, the number of pro-inflammatory immune cells is greatly elevated [95]. A struggle takes place between the pro- and anti-inflammatory components in innate and adaptive immunity. Macrophages, neutrophils, and mast cells are the major components of innate immunity [96]. Specifically, basophils, which are associated with T helper 2 (Th2) immune responses, have the capacity to initiate and expand inflammation through the production of specific cytokines and proteases that lead to inflammation [97]. In addition, tissue mast cells are very similar to circulating basophil granulocytes; both contain histamine, serotonin, and heparin. High serotonin levels are associated with obesity, and a deficit in serotonin has been shown to prevent the development of obesity and insulin resistance [98]. This therefore raises a question as to the pathophysiological role of quantitative or qualitative regulation of serotonin production in lipomatosis, especially in the painful lipomas of DD. 

## 5. Conclusions

Lipomatosis is a rare disease, and most types are considered benign despite the consequences of pain on the quality of life in DD. The diseases remain underdiagnosed [99]. In addition to multiple lipomas, which are painful in DD, both DD and RLML evolve in association with a metabolic syndrome, which is characterized by the presence of overweight, hypertension, diabetes, or dyslipidemia. The long-term course remains poorly described. The pathophysiology is unknown but could be related to obesity and low-grade inflammation. 

## Figures and Tables

**Figure 1 jcm-10-01292-f001:**
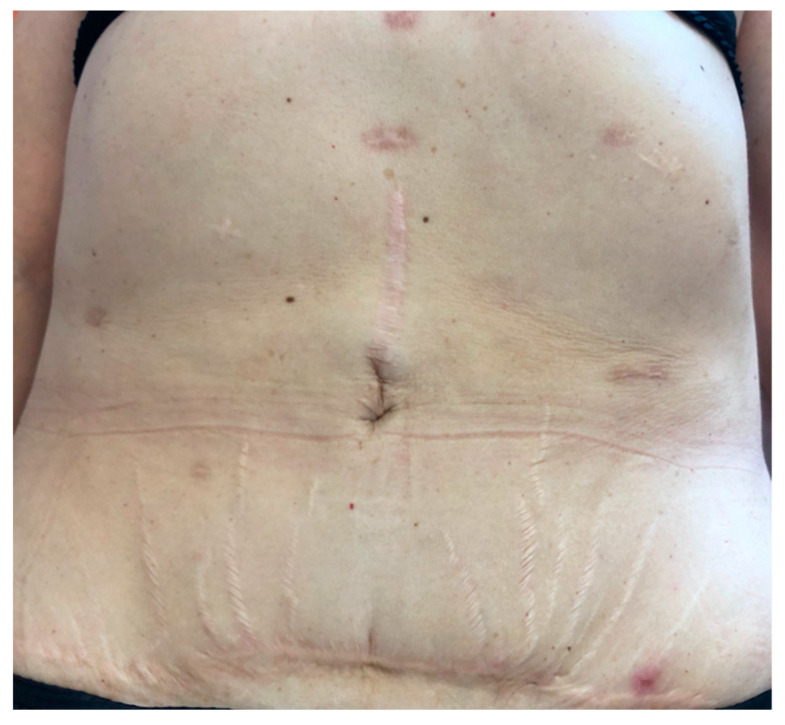
Dercum’s lipomatosis: Multicicatricial abdomen linked to multiple excisions and multiple encapsulated lipomas, associated with pain in a woman with Dercum’s disease progressing in a context of obesity.

**Figure 2 jcm-10-01292-f002:**
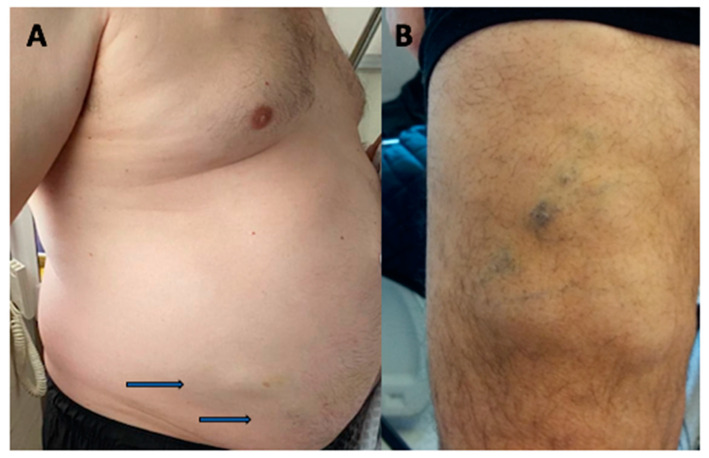
Roch-Leri mesosomatous lipomatosis. (**A**) Lipomas distributed over the abdomen, painless and non-tender to palpation, with progressive onset of several years. (**B**) Multiple encapsulated lipomas, painless, disseminated on the thighs with an ecchymotic tendency.

**Table 1 jcm-10-01292-t001:** Lipomatosis: Lipomatosis can be isolated or be part of a syndrome.

Isolated Lipomatosis	Syndromic Lipomatosis
Familial multiple lipomatosis	Proteus syndrome
Multiple symmetric lipomatosis	Cowden syndrome
Hibernomas	MERRF syndrome
Epidural lipomatosis	CLOVES syndrome
Familial angiolipomatosis	HHML syndrome
Dercum’s disease	SOLAMEN syndrome
Roch-Leri mesosomatous lipomatosis	

MERRF syndrome: myoclonic epilepsy with red ragged fibers; CLOVES syndrome: congenital, lipomatous, overgrowth, vascular malformation, epidermal nevi, scoliosis, skeletal, spinal anomalies; HHML syndrome: hemihyperplasia–multiple lipomatosis; SOLAMEN syndrome: segmental overgrowth, lipomatosis, arteriovenous malformation, and epidermal nevus.

**Table 2 jcm-10-01292-t002:** Histological data from the literature concerning Dercum’s disease (DD) and Roch-Leri mesosomatous lipomatosis [37,42,62,63,64]. IL: interleukin.

Study	Year	Number of DD Patients	Methods	Number and Type of Control Subjects	Results
Adiposis dolorosa: Report of a case with increased sugar tolerance and epileptiform convulsions. Price GE. and Bird JT [62]	1905	2	Necropsy	-	Fibrous component Some granulomas, angiomas, or vascular thromboses
Adiposis dolorosa (Dercum’s disease). Steiger WA [63]	1952	1	Fat biopsy		Increased proportion of connective tissue
Fat-cell heat production, adipose tissue fatty acids, lipoprotein lipase activity and plasma lipoproteins in adiposis dolorosa. Fagher B [37]	1991	13	Buttock biopsy in a painful area	Weight-matched subjects (*n* = 27)	Larger fat cells
				Severely obese group (*n* = 14)	Similar fat cell size Higher cellular heat production suggesting possible sympathetic damage
Lipomatosis-associated inflammation and excess collagen may contribute to lower relative resting energy expenditure in women with adiposis dolorosa. Herbst KL [42]	2009	5	Fat biopsy in painful areas	Healthy control group (*n* = 5)	Higher level of IL6 No difference for expressions levels of TNF-alpha, IL-13, IL-8, IL-1B Multi-nucleated giant cells (3/5 DD vs. 0/5 in the control group). Same macrophage count
Histology of adipose tissue inflammation in Dercum’s disease, obesity and normal weight controls: A case control study. Hansson E [64]	2011	53	Fat biopsy in painful areas (abdomen or knee)	Obese control group (*n* = 41)	Same inflammatory response as the obese control group
				Non-obese control group (*n* = 11)	Higher inflammatory response than in the non-obese control group

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
