# Peer review of "Rare Forms of Lipomatosis: Dercum’s Disease and Roch-Leri Mesosomatous Lipomatosis"

_jcm, 2021, doi:10.3390/jcm10061292_

Round 1

Reviewer 1 Report

Dear Authors, 

although the subject matter has a rare incidence, I find it to have a considerable clinical and practical value.
I think your article is well structured, despite having identified some critical issues.
I find that the paragraph “Therapeutic Management of Decrum’s disease” lack of fundamental scientific content highlighted in the last few years. The “Surgical Aspect of Decrum’s disease” could be better deepened, differentiating it according to the 4 subtypes. 
In “Other therapies reported in the literature of Decrum’s disease” the list of treatments could be improved, dividing it into sub-clinical (pharmacological therapies, conservative non-surgical therapies, surgical therapies). 
Finally, it is necessary to review the bibliography which is not well structured.

Author Response

  • I find that the paragraph “Therapeutic Management of Decrum’s disease” lack of fundamental scientific content highlighted in the last few years.
  • We have completed the review of therapeutic management (underlined in yellow) and added a reference: AdiposisDolorosa Pain Management. Eliason AH, Seo YI, Murphy D, Beal C.Fed Pract. 2019 Nov;36(11):529-533

  • The “Surgical Aspect of Dercum’s disease” could be better deepened, differentiating it according to the 4 subtypes.
  • We thank the reviewer for this comment and have specified the indications of each type of surgery in each of the 4 types of DD.

  • In “Other therapies reported in the literature of Dercum’s disease” the list of treatments could be improved, dividing it into sub-clinical (pharmacological therapies, conservative non-surgical therapies, surgical therapies):
  • We have followed your advice and have amended this paragraph accordingly.

  • Finally, it is necessary to review the bibliography which is not well structured.
  • We have modified the references according to the guidelines available

Reviewer 2 Report

There is some confusion in the Dercum disease definition. The authors consider that lipomas should be always present, however this is not always the case. In fact, the classification includes Type 1 (generalized diffuse), in which lipomas are not present.

Please, describe the clinical features of each Dercum diseases subtype. 

Regarding DD treatment, writing just "antidepressant?", it sound a little odd. Probably , a something  "more extended" comment should be necessary.

References must follow the Journal rules (e.g. including DOI).

Author Response

  • There is some confusion in the Dercum disease definition. The authors consider that lipomas should be always present, however this is not always the case. In fact, the classification includes Type 1 (generalized diffuse), in which lipomas are not present.Please, describe the clinical features of each Dercum diseases subtype. 

      We had just mentioned the different forms of Dercum; we have therefore detailed them in paragraph “2.3 symptomatology”, in accordance with your suggestion.

  • Regarding DD treatment, writing just "antidepressant?", it sounds a little odd. Probably, a something  "more extended" comment should be necessary.

The antidepressants reported in the literature are essentially modulators of the calcium chain, which are also cited in the section "12. Other therapies reported in the literature (only in case reports): ». We have therefore deleted this term from paragraph « 2.9. Psychological support » and have moved it to “analgesics”

  • References must follow the Journal rules (e.g. including DOI).

References were modified according to the guidelines available. The DOI was not required but we added it

Note that the numbering of sections was wrong, and we corrected it
